# On Evaluation of Bangla Word Analogies

**Mousumi Akter, Souvika Sarkar, Shubhra Kanti Karmaker ("Santu")**
Big Data Intelligence (BDI) Lab
Department of Computer Science & Software Engineering
Auburn University, Alabama, USA
{mza0170, szs0239, sks0086}@auburn.edu

## Abstract

This paper presents a benchmark dataset of Bangla word analogies for evaluating the quality of existing Bangla word embeddings. Despite being the 7th largest spoken language in the world, Bangla is still a low-resource language and popular NLP models often struggle to perform well on Bangla data sets. Therefore, developing a robust evaluation set is crucial for benchmarking and guiding future research on improving Bangla word embeddings, which is currently missing. To address this issue, we introduce a new evaluation set of 16,678 unique word analogies in Bangla as well as a translated and curated version of the original Mikolov dataset (10,594 samples) in Bangla. Our experiments with different state-of-the-art embedding models reveal that current Bangla word embeddings struggle to achieve high accuracy on both data sets, demonstrating a significant gap in multilingual NLP research.

## 1 Introduction

The Bangla language, having over 300 million native speakers and ranking as the seventh most spoken language in the world, is still regarded as a language with limited resources (Joshi et al., 2020). Despite the breakthrough in the field of Natural Language Processing (NLP) recently, it has been reported that popular NLP models often fail to perform well for low-resource languages like Bangla while showing human-like performance in high-resource languages like English. As such, there is a growing need for high-quality Bangla word embeddings, and several efforts have been made in this direction by training embeddings on large Bangla corpora (Bhattacharjee et al., 2022a; Artetxe and Schwenk, 2019; Feng et al., 2022). For evaluation, there have been a few efforts to create Bangla benchmark data sets for downstream NLP tasks like Sentiment Analysis, Machine Translation, and Summarization (Hasan et al., 2020, 2021; Akil et al., 2022; Bhattacharjee et al., 2022b). On the contrary, there is a clear absence of word analogy data sets for evaluating the syntactic and semantic properties of Bangla word embeddings, although such word analogy evaluation data sets are available for other languages Like Turkish, German, Spanish, Arabic, etc. (Gurevych, 2005; Hassan and Mihalcea, 2009; Sak et al., 2010; Joubarne and Inkpen, 2011; Panchenko et al., 2016).

As word embeddings have direct impacts on the performance of downstream NLP tasks, creating a high-quality evaluation test set for Bangla word embeddings will enable researchers to benchmark the performance of existing Bangla embedding models and guide further research. With this motivation, we present a Mikolov-style (Mikolov et al., 2013) high-quality word-analogy evaluation set exclusively for Bangla, with a sample size of 16,678[1]. To the best of our knowledge, we are the first ones to do so. In Addition, we translated and curated Mikolov's original dataset for Bangla, resulting in 10,594 more analogies. For both these data sets, we also provide an analysis of the performance of several state-of-the-art Bangla/multilingual embedding models: Word2Vec (Mikolov et al., 2013), GloVe (Pennington et al., 2014), fastText (Bojanowski et al., 2017), LASER (Artetxe and Schwenk, 2019), LaBSE (Feng et al., 2022), bnBERT (Bhattacharjee et al., 2022a), bnBART (Wolf et al., 2020).

Our experiments with different state-of-the-art embedding models reveal that current Bangla word embeddings struggle to achieve high accuracy on both data sets, signifying that Bangla has its own unique characteristics, and further research is warranted for low-resource languages like Bangla. Therefore, we suggest that future research on Bangla word embeddings should report accuracy using this new benchmark data set in order to track consistent research progress in this direction.

---

[1]https://paperswithcode.com/dataset/bangla-word-analogy

| Type | Relationship | Subgroup | Sample # | Example Word Pair 1 | | Example Word Pair 2 | |
|---|---|---|---|---|---|---|---|
| Semantic | Division-District | | 2160 | ঢাকা (Dhaka) | শরীয়তপুর (Shariatpur) | রংপুর (Rangpur) | কুড়িগ্রাম (Kurigram) |
| | Gender | | 1260 | বাবা (Father) | মা (Mother) | চাচা (Uncle) | চাচী (Aunt) |
| | Number | Ordinal | 380 | এক (One) | প্রথম (First) | পাঁচ (Five) | পঞ্চম (Fifth) |
| | | Date | 930 | এক (One) | পহেলা (First) | দুই (Two) | দোসরা (Second) |
| | | Female | 306 | এক (One) | প্রথমা (First) | দুই (Two) | দ্বিতীয়া (Second) |
| Syntactic | Comparative | | 552 | শীতল (Cool) | শীতলতর (Cooler) | দীর্ঘ (Long) | দীর্ঘতর (Longer) |
| | Superlative | | 600 | দীর্ঘ (Long) | দীর্ঘতম (Longest) | নিম্ন (Low) | সর্বনিম্ন (Lowest) |
| | Antonym | Adjective | 4692 | দামী (Expensive) | সস্তা (Cheap) | আসক্ত (Addicted) | নিরাসক্ত (Desperate) |
| | | Misc. | 3782 | আগমন (Arrival) | প্রস্থান (Departure) | উজান (Upstream) | ভাটি (Downstream) |
| | Plural | Noun | 506 | মাঝি (Sailor) | মাঝিরা (Sailors) | ছাত্র (Student) | ছাত্ররা (Students) |
| | | Object | 72 | রচনা (Composition) | রচনাবলি (Compositions) | নিয়ম (Rule) | নিয়মাবলি (Rules) |
| | Tense | | 144 | চলছি (Continued) | চলছ (Continuing) | চলব (Will continue) | চলবে (Will Continue) |
| | Prefix | | 66 | জ্ঞান (Knowledge) | বিজ্ঞান (Science) | শেষ (End) | বিশেষ (Special) |
| | Suffix | | 94 | ভাব (Attitude) | ভাবখানা (Attitude) | ব্যাপার (Matter) | ব্যাপারখানা (Matter) |
| | Affix | | 38 | রাত্রি (Night) | রাত্রিতে (Night) | হাতি (Elephant) | হাতিতে (Elephant) |
| | Standard - Colloquial | Noun | 342 | মৎস্য (Fish) | মাছ (Fish) | হস্তী (Elephant) | হাতি (Elephant) |
| | | Pronoun | 110 | তাহাকে (His) | তাকে (His) | তাহার (His) | তার (His) |
| | | Verb | 462 | হইলাম (Become) | হলাম (Become) | করিবার (Do) | করার (Do) |
| | | Conjunction | 182 | যদ্যপি (Although) | যদিও (Although) | প্রায়শ (Often) | প্রায়ই (Often) |

Table 1: This table shows the statistics of the dataset and examples of the semantic and syntactic relationship sets. Relationships highlighted in cyan represent unique linguistic forms for Bangla, while highlighted in red represent somewhat unique forms with different syntax compared to English.

## 2 Dataset

To evaluate the quality of word vectors, the authors (who are also native Bengali speakers) independently first proposed different types of relationships for the Bangla Language. Subsequently, they created a list of related word pairs for each relationship and then formed analogies by grouping two pairs. The pairs were then independently reviewed by other annotators and removed if there was any disapproval from the annotator. Thus, a comprehensive test set was developed that contains three types of semantic analogies and nine types of syntactic analogies, as shown in Table 1. Overall, our testing set contains 5,036 semantic and 11,642 syntactic analogies, providing a comprehensive collection for evaluating the quality of Bangla word vectors. For example, we made a list of 7 divisions in Bangladesh and 64 districts that belong to these divisions and formed 2,160 analogies by picking every possible division-district pair. Following Mikolov's approach (Mikolov et al., 2013), word analogies were created by using the format $word_B$ - $word_A$ = $word_D$ - $word_C$. The goal was to determine the $word_D$ that is similar to $word_C$ in the same way that $word_B$ is similar to $word_A$.

While creating this test set, we also took into account the unique characteristics of the Bangla language. For example, the Bangla language has different forms for numbers, including date forms, female forms, and also with prefixes and suffixes that change the meaning of the word. Additionally, Bangla has colloquial forms that are often used in literature, stories, and novels. A total of 2,844 word pairs were formed that reflect these unique characteristics of the Bangla language. Furthermore, 3,776 word pairs were introduced in the evaluation set that is absent in Mikolov's word analogy data set, such as division-district pairs and number pairs with different forms. These additional word forms further demonstrate the diverse and complex nature of the Bangla language.

Additionally, we translated Mikolov's original dataset[2] and manually removed English words that do not have Bangla translations. We also removed word pairs that translate to duplicated terms in Bangla, such as present participles and plural verb forms. This cleaning step resulted in a dataset of 10,594 samples from the original 19,544 samples. In summary, while our new dataset focused on the linguistic specifics of the Bangla language, the translated Mikolov's dataset is more focused on common words in both Bangla and English. The translation of Mikolov's dataset provides a useful resource for cross-lingual research and analysis.

## 3 Experimental Setup

We evaluated the quality of pretrained word vectors trained using both traditional and transformer-based models on both data sets. For classical models, we used Word2Vec, GloVe, and fastText, while for transformer-based models, we employed LASER, LaBSE, bnBERT, and bnBART, to evaluate the quality of word embeddings.

[2]www.fit.vutbr.cz/ imikolov/rnnlm/word-test.v1.txt

| Embedding | Word2Vec | GloVe | fastText | LaBSE | bnBERT | LASER | bnBART |
|-----------|----------|-------|----------|-------|--------|-------|--------|
| **Dimension** | 100 | 100 | 300 | 768 | 768 | 1024 | 1024 |

Table 2: Dimensions of different embedding used

For GloVe, we used the Bengali GloVe model that was trained on Bangla Wikipedia[3] and Bangla news articles. Specifically, we utilized the model trained with 39M tokens. For Word2Vec, we used the Bengali Word2Vec model, also trained with Bengali Wikipedia data (Sarker, 2021). We also used fastText, which provides pretrained Bangla word embeddings (Bojanowski et al., 2017) .

For transformer-based models, we used LASER (Artetxe and Schwenk, 2019), LaBSE (Feng et al., 2022), bnBERT (Bhattacharjee et al., 2022a), bnBART (Wolf et al., 2020), which provide sentence embeddings. To obtain the word embedding for a particular word, we passed the word to the model and collected the contextualized token embeddings for all the tokens of that word, which were then averaged to obtain the word embedding. This way, we created an exhaustive embedding dictionary for 178,152 Bengali words and used them directly to perform the word analogy task. For experiments, we used NVIDIA Quadro RTX 5000 GPUs. Table 2 shows the dimensions of different embedding used. In summary, we performed a comprehensive evaluation of pretrained word vectors (using both traditional and transformer-based embedding models) on our two data sets, following the same word analogy task setup used by Mikolov et al. (2013).

### 3.1 Bangla Word Embedding

**Word2Vec**[4]: The Wikipedia dump datasets were used for training Word2Vec word embeddings of Bangla with a dimension of 100. The minimum count of words was set to 5, the window size was 5, and the training was performed over 10 epochs.
**GloVe**[5]: The Bangla GloVe Vectors offer pretrained glove vectors for the Bengali language. The model is Trained using Wikipedia dump datasets and crawled bangla news articles with specific parameters including a vector size of 100, a window size of 15, and a maximum iteration of 15. With a vocabulary size of 0.18 million.
**fastText**[6]: The fastText pre-trained model for the

Bengali language was trained on Wikipedia dump datasets. It consists of 20 million words, a vocabulary size of 1171011, and was trained for 50 epochs with a 300-dimensional embedding.
**LASER**[7]: LASER model performs zero-shot cross-lingual transfer across 90+ languages. It is open-source and supports efficient language processing and understanding.
**LaBSE**[8]: LaBSE is a BERT-based model trained on 109 languages for sentence embedding, combining masked language modeling and translation language modeling. It provides multilingual sentence embeddings.
**bnBERT**[9]: BanglaBERT is an ELECTRA discriminator model pretrained with the Replaced Token Detection (RTD) objective, suitable for finetuning on diverse downstream tasks in Bengali.
**bnBART**[10]: bnTransformer is built with transformers for various transformer-based inference tasks in Bengali. We pass *facebook/bart-large-cnn model* from the huggingface model hub to get the embeddings.

## 4 Results

Our first set of results is focused on our handcrafted Bangla analogy set (presented in Table 1). In all our experiments, we strictly matched against the target word from our dataset to compute Top-1, Top-3, Top-5, and Top-10 accuracy numbers. Because of strict matching, achieving a very high accuracy was not expected. However, as demonstrated in Figure 1, the accuracy numbers turned out to be even worse than expected. For example, the best overall top-5 accuracy was obtained by bnBART (a 1024-dimensional embedding), which turned out to be only 20.9%. LaBSE (a 768-dimensional embedding) yielded the best Top-5 accuracy for semantic analogies (around 12%), while bnBART gave the best Top-5 syntactic accuracy (around 25%). Top-1 accuracy was generally low in all cases, except for bnBART achieving around 21% Top-1 accuracy for syntactic analogies.

Next, we compare our handcrafted dataset

---

[3] bn.wikipedia.org/
[4] huggingface.co/sagorsarker/bangla_word2vec
[5] huggingface.co/sagorsarker/bangla-glove-vectors
[6] github.com/facebookresearch/fastText/

[7] github.com/facebookresearch/LASER
[8] huggingface.co/sentence-transformers/LaBSE
[9] huggingface.co/csebuetnlp/banglabert
[10] github.com/sagorbrur/bntransformer

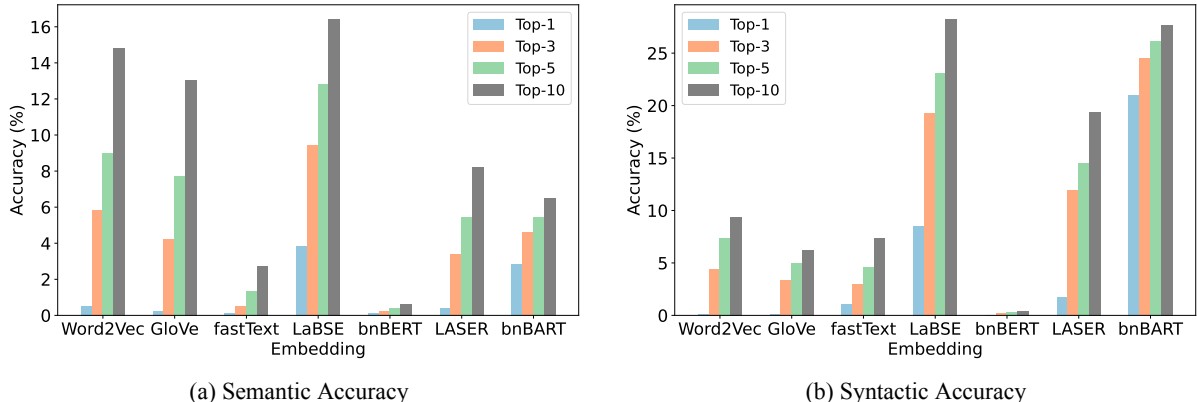

(a) Semantic Accuracy

(b) Syntactic Accuracy

Figure 1: Top-1(%), Top-3(%), Top-5(%) and Top-10(%) accuracy for different embeddings on our dataset.

| Type | Relationship | Top 5% Performance Break-Down for Different Categories | | | | | | |
|---|---|---|---|---|---|---|---|---|
| | | Word2Vec | GloVe | fastText | LaBSE | bnBERT | LASER | bnBART |
| Semantic | Division-District | **9.8** | 9.5 | 0.0 | 0.1 | 0.6 | 0.0 | 0.0 |
| | Gender | 11.2 | 8.5 | 3.3 | **21.9** | 0.1 | 12.7 | 12.6 |
| | Number | 5.9 | 5.1 | 0.6 | **16.3** | 0.4 | 3.4 | 3.7 |
| Syntactic | Antonym | 11.0 | 7.6 | **14.5** | 3.5 | 0.7 | 1.0 | 3.3 |
| | Tense | 6.3 | 2.8 | 4.2 | 13.9 | 0.0 | **16.7** | 14.6 |
| | Comparative | 5.1 | 1.1 | 6.5 | 12.5 | 2.2 | 5.3 | **14.7** |
| | Superlative | 3.5 | 1.3 | 3.3 | **10.2** | 0.0 | 6.3 | 5.2 |
| | Prefix | 1.5 | 0.0 | 3.0 | 9.1 | 0.0 | 3.0 | **43.9** |
| | Suffix | 0.0 | 0.8 | 0.8 | 30.4 | 0.0 | 22.8 | **53.6** |
| | Affix | 21.1 | 23.7 | 5.3 | 63.2 | 0.0 | 50.0 | **79.0** |
| | Plural | 7.3 | 2.5 | 1.5 | **41.6** | 0.0 | 18.4 | 11.1 |
| | Standard-Colloquial | 9.8 | 3.9 | 1.9 | **23.5** | 0.0 | 6.6 | 9.4 |
| | **Overall Accuracy** | 7.7 | 5.6 | 3.7 | 20.5 | 0.3 | 12.2 | 20.9 |
| Type | Relationship | Top 10% Performance Break-Down for Different Categories | | | | | | |
| | | Word2Vec | GloVe | fastText | LaBSE | bnBERT | LASER | bnBART |
| Semantic | Division-District | **21.5** | 17.5 | 0.0 | 0.1 | 0.7 | 0.0 | 0.0 |
| | Gender | 13.3 | 11.6 | 6.4 | **27.5** | 0.1 | 19.4 | 14.3 |
| | Number | 9.6 | 9.8 | 1.7 | **21.7** | 1.1 | 5.3 | 5.2 |
| Syntactic | Antonym | 14.6 | 10.5 | **22.6** | 5.3 | 0.8 | 1.6 | 4.7 |
| | Tense | 9.7 | 3.5 | 5.6 | 16.0 | 0.0 | **20.1** | 18.8 |
| | Comparative | 6.2 | 2.2 | 10.0 | 13.0 | 3.1 | 8.3 | **15.2** |
| | Superlative | 5.3 | 1.3 | 5.5 | **12.5** | 0.0 | 8.3 | 5.8 |
| | Prefix | 1.5 | 1.5 | 6.1 | 10.6 | 0.0 | 4.6 | **43.9** |
| | Suffix | 0.0 | 0.8 | 1.7 | 37.3 | 0.0 | 31.2 | **53.6** |
| | Affix | 21.1 | 26.3 | 7.9 | 73.7 | 0.0 | 63.2 | **81.6** |
| | Plural | 11.3 | 3.7 | 2.6 | **53.8** | 0.0 | 27.7 | 14.6 |
| | Standard-Colloquial | 13.6 | 5.9 | 3.3 | **31.2** | 0.0 | 9.0 | 10.1 |
| | **Overall Accuracy** | 10.6 | 7.9 | 6.1 | 25.2 | 0.5 | 16.6 | 22.3 |

Table 3: Top-5(%) and 10 (%) accuracy on three types of semantic and nine types of syntactic relationship set. The highest accuracy for each category is bolded. Results indicate that certain embeddings perform strongly, moderately strongly, or weakly depending on the relationship type, with green, blue, and red highlights, respectively.

against the translated Mikolov dataset in Figure 2 and 3. Again, LaBSE and bnBART were the overall winners, and both methods performed better on the handcrafted dataset than the translated one.

## 4.1 Discussion

An interesting observation from the results is that training embeddings on multilingual corpora and multiple downstream tasks helped models better capture the syntactic relations among Bangla words. For example, both LaBSE and LASER, which achieved high accuracy on syntactic analogies, were trained with multilingual data, where LaBSE was trained on data from 109 languages, and LASER was trained on data from 97 lan-

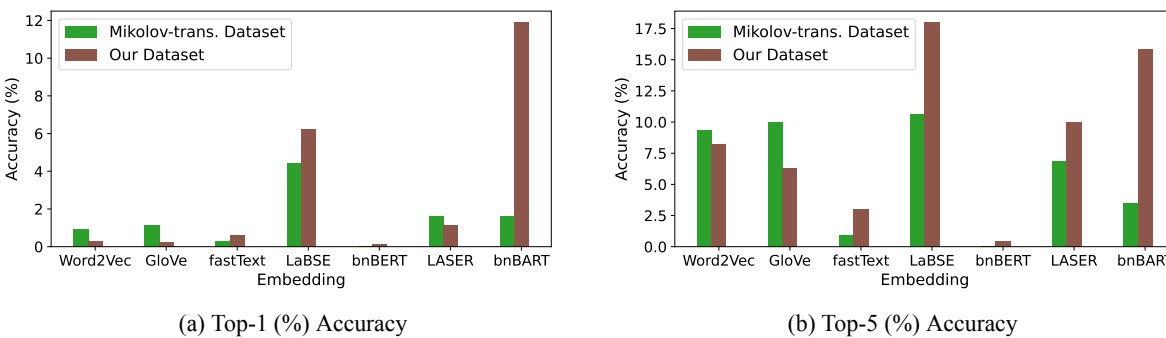

(a) Top-1 (%) Accuracy           (b) Top-5 (%) Accuracy

Figure 2: Comparison of Top-1(%) and Top-5(%) accuracy on translated Mikolov dataset and our dataset.

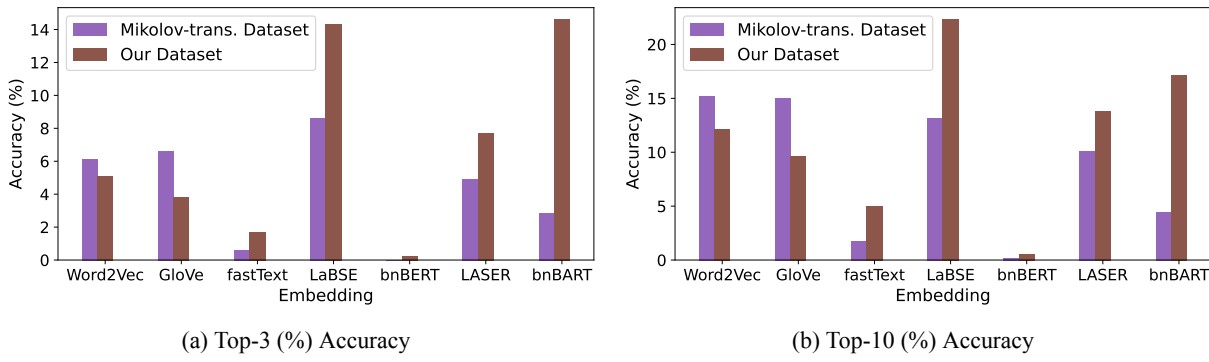

(a) Top-3 (%) Accuracy           (b) Top-10 (%) Accuracy

Figure 3: Comparison of Top-3 (%) and Top-10 (%) accuracy on translated Mikolov dataset and our dataset .

guages (Feng et al., 2022). On the other hand, bnBART, another high-performing embedding on syntactic analogies, was finetuned on multiple downstream inference tasks for the Bengali language, such as QA, NER, MASK generation, Translation, and Text generation. In contrast, Word2Vec, GloVe, and fastText achieved lower accuracy on syntactic analogies and were trained on only Bangla Wikipedia and crawled Bangla news articles.

For semantic analogies, LaBSE performed the best, followed by Word2Vec. Table 3 demonstrates the category-wise performance in terms of Top-5 and Top-10 accuracy, where LaBSE and bnBART turned out to be the overall winners. However, with 20.9% and 25.2% best Top-5 and Top-10 accuracy scores, respectively, we can safely conclude that state-of-the-art Bangla/multilingual embeddings struggle a lot to capture the semantic and syntactic relations among Bangla words properly.

## 5 Conclusion

In this paper, we have introduced a high-quality dataset for evaluating Bangla word embeddings through word analogy tasks, similar to Mikolov's word analogy data set for English. Additionally, we translated Mikolov's original dataset into Bangla, contributing a second data set. Our exper-

iments with different word embeddings reveal that current word embeddings for Bangla still struggle to achieve high accuracy on both data sets. To improve performance, future research should focus on training embeddings with larger data sets while taking into account the unique morphological characteristics of Bangla.

## 6 Limitations

It is important to acknowledge that our study has certain limitations. Firstly, we tried to provide an exhaustive list of all possible word analogies from the Bangla language. However, it is possible that we may unintentionally miss some analogies and the data sets can be extended by adding those analogies. Additionally, we created the translated data set from English word analogies only, and we did not consider other languages. Future research can be conducted with the translated dataset from other languages to explore the potential differences in word analogy relations across languages. Moreover, the evaluation of word analogy with proprietary large language models, such as ChatGPT and Bard, was not explored in this study. Rather, the focus was entirely on models from which Word embeddings could be directly obtained and subsequently tested. Further research may be undertaken in this direction with large language models.

# 7 Acknowledgements

This work has been partially supported by the National Science Foundation (NSF) Standard Grant Award #2302974 and Air Force Office of Scientific Research Grant/Cooperative Agreement Award #FA9550-23-1-0426. We would also like to thank Auburn University College of Engineering and the Department of CSSE for their continuous support through Student Fellowships and Faculty Startup Grants.

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
