# OpenReview forum: "On Evaluation of Bangla Word Analogies"
_EMNLP/2023/Conference — EMNLP 2023 Main_

### Official Review · Reviewer_Wxve · 2023-07-21

**Soundness:** 4

**Excitement:**

3: Ambivalent: It has merits (e.g., it reports state-of-the-art results, the idea is nice), but there are key weaknesses (e.g., it describes incremental work), and it can significantly benefit from another round of revision. However, I won't object to accepting it if my co-reviewers champion it.

**Paper Topic And Main Contributions:**

This paper introduces two word analogy datasets for the low-resource Bangla
language: i) by translating the existing English resource (Mikolov et al., 2013)
and ii) building a dataset from scratch, keeping Bangla linguistic properties in
mind. The authors evaluate various classical, such as Word2Vec, and transformers
based word embedding models, and find that the latter group of models
outperforms the former and that the overall performance of the models is low,
indicating that further work is needed to improve Bangla word representation
quality. The new dataset, which is the main contribution of the paper, is an
important resource to help the evaluation of future methods on the Bangla
language.


**Questions For The Authors:**

* See reasons to reject section.


**Reasons To Accept:**

* The paper introduces a new evaluation dataset for the low-resource Bangla
language which will support other researches in the development of NLP tools for
the language.
* Various word representation methods are evaluated and compared, including
their fine-grained evaluation on the 12 word analogy relationship types.
* The paper is well written and easy follow.


**Reasons To Reject:**

* A: The description of how the new dataset was built (first half of section 2) is
a bit brief, some details are missing. For example, how were the list of related
word pairs acquired, how were two pairs selected to form a group (while the
mentioned division-district example is helpful, other groups are not clear), how
many annotators (if any) and with what background worked on the dataset,
agreement rate, etc.? Additionally, some of the Bangla specific examples in
table 1 are not clear for non Bangla speakers, e.g., how are the two words in
the Fish-Fish pair different? These specifics should be explained in more
detail.
* B: The authors mention a few already existing resources for Bangla which are
mainly related to sentence or document level tasks, such as sentiment
classification or summarization. This paper made me wonder about the importance
of word level tasks, such as analogy, which is not discussed in the paper.
However, I think it would be important to motivate the importance of the task,
since most recent NLP work focus on large language models and sentence/document
level tasks.


**Reproducibility:**

4: Could mostly reproduce the results, but there may be some variation because of sample variance or minor variations in their interpretation of the protocol or method.

**Reviewer Confidence:**

4: Quite sure. I tried to check the important points carefully. It's unlikely, though conceivable, that I missed something that should affect my ratings.

---

> ### Author Rebuttal · Authors · 2023-08-26
>
> - “The description of how the new dataset was built (first half of section 2) is a bit brief, some details are missing. For example, how were the list of related word pairs acquired, how were two pairs selected to form a group (while the mentioned division-district example is helpful, other groups are not clear), how many annotators (if any) and with what background worked on the dataset, agreement rate, etc.? Additionally, some of the Bangla specific examples in table 1 are not clear for non Bangla speakers, e.g., how are the two words in the Fish-Fish pair different? These specifics should be explained in more detail.”
>
>
>   - Thank you for your comment. Yes, you are right, the highlighted pairs in cyan are hard to distinguish for non-native Bengali speakers, but they are unique to the Bangla language, making our dataset unique for research in Bangla. Due to space limitations, we could not provide more details on data-set creation in the main paper and would be happy to include them in the extra page/appendix in the camera-ready version (if accepted). For now, 3 native Bengali speakers who are also NLP researchers independently first manually proposed different types of relationships for the Bangla Language. Subsequently, they created a list of related word pairs for each relationship. Next, they checked the pairs created by other annotators. If other annotators disagree, we dropped that word pair. Finally, the two words in the Fish-Fish pair are different because one of them is the standard Bangla form (mostly used in formal writing), and the other one is the Colloquial form (used for daily speaking purposes).
>
> - “The authors mention a few already existing resources for Bangla which are mainly related to sentence or document level tasks, such as sentiment classification or summarization. This paper made me wonder about the importance of word level tasks, such as analogy, which is not discussed in the paper. However, I think it would be important to motivate the importance of the task, since most recent NLP work focus on large language models and sentence/document level tasks.”
>
>   - This is an excellent comment and, indeed, worth discussing. Note that a word is still the foundation of any language, and word embeddings directly impact the performance of downstream NLP tasks. Although contextual embeddings and in-context learning capabilities are the central focus of NLP now, those contextual embeddings are usually derived from nonlinear transformations of context-free (classic) embeddings and hence, improving classic context-free embeddings will directly/indirectly improve contextual embeddings and in-context learning capabilities, which in turn, will also enable improvement in sentence/document-level and downstream tasks.

---

### Official Review · Reviewer_5yeX · 2023-08-03

**Soundness:** 4

**Excitement:**

3: Ambivalent: It has merits (e.g., it reports state-of-the-art results, the idea is nice), but there are key weaknesses (e.g., it describes incremental work), and it can significantly benefit from another round of revision. However, I won't object to accepting it if my co-reviewers champion it.

**Paper Topic And Main Contributions:**

In this paper, the authors create a new word analogy dataset for Bangla. A diverse set of relations are covered spanning many synactic and semantic relation categories. They also translate and filter the original Mikolov dataset. They compare various models (classical word embedding models and sentence embeddings models) on this dataset.


**Questions For The Authors:**

- When the dataset is released, please retain mappings to the original Mikolov dataset.
- Line 139: Are you using an existing Glove embedding or have you trained one?
- Given that the NLP world is now focused on contextual embeddings and in-context learning capabilities, the utility of a dataset studying linear transformations of word embeddings seems limited. A comment on the current relevance of the dataset would be helpful.

**Reasons To Accept:**

- Creation of a word analogy dataset for Bangla that is diverse.
- Translation of an English word analogy dataset can help cross-lingual studies.


**Reasons To Reject:**

- There is limited learning from the paper about word analogy in Bangla beyond reporting accuracies on a few pre-trained embeddings. It would have been interesting to answer questions like: (a) how does Bangla word analogy compare with English word analogy results? This is possible since the authors have also translated Mikolov's original dataset (b) If we use a model that is also trained on related languages (like IndicBERT or IndicBART) would that improve model performance?
- Details about the bnBART model are missing. The citation (line 71) is wrong. It is not clear what dataset this model is trained on. Line 202 seems to suggest that this model is trained on many NLP tasks - however, there is no paper mentioning this and no information in the git repo.

**Reproducibility:**

4: Could mostly reproduce the results, but there may be some variation because of sample variance or minor variations in their interpretation of the protocol or method.

**Reviewer Confidence:**

4: Quite sure. I tried to check the important points carefully. It's unlikely, though conceivable, that I missed something that should affect my ratings.

---

> ### Author Rebuttal · Authors · 2023-08-26
>
> - “There is limited learning from the paper about word analogy in Bangla beyond reporting accuracies on a few pre-trained embeddings. It would have been interesting to answer questions like: (a) how does Bangla word analogy compare with English word analogy results? This is possible since the authors have also translated Mikolov's original dataset (b) If we use a model that is also trained on related languages (like IndicBERT or IndicBART) would that improve model performance?”
>
>     - Regarding a) We have shown the comparison of the Bangla word analogy with the translated English word analogy in Figure 2, and we are not sure what other comparisons we are missing here. Could you please provide more clarification here? Regarding b) We appreciate your comment and will be happy to include more models specifically trained on related languages (like IndicBERT or IndicBART) in the camera-ready version (if accepted).
>
>
> - “Details about the bnBART model are missing. The citation (line 71) is wrong. It is not clear what dataset this model is trained on. Line 202 seems to suggest that this model is trained on many NLP tasks - however, there is no paper mentioning this and no information in the git repo.”
>
>     - Thank you for this thorough and specific comment. The exact bnBART model we have reported is from https://github.com/sagorbrur/bntransformer/ (line 202). This model has been pretrained on different inference tasks for the Bangla language, and we have used the BART model from Huggingface (https://github.com/huggingface/transformers) as a tokenizer. Unfortunately, no paper is available to cite on the GitHub link; therefore, we have cited the paper from here (https://github.com/huggingface/transformers) and provided the link to the original trained model on Bengali (line 202). Regarding the concern of the trained dataset, it’s included at the bottom of the GitHub link, also including it here.
>       - Question Answering (https://huggingface.co/sagorsarker/mbert-bengali-tydiqa-qa)
>       - Name Entity Recognition (https://huggingface.co/neuropark/sahajBERT-NER)
>       - Mask Generation (https://huggingface.co/sagorsarker/bangla-bert-base)
>       - Translation (https://huggingface.co/Helsinki-NLP/opus-mt-bn-en)
>       - Text Generation (https://huggingface.co/flax-community/gpt2-bengali)
>
> - “Are you using an existing Glove embedding, or have you trained one?”
>
>     - We are using pretrained Glove embedding, which was trained on Bangla Wikipedia datasets and crawled Bangla news articles. The details are provided in the appendix.
>
> - “Given that the NLP world is now focused on contextual embeddings and in-context learning capabilities, the utility of a dataset studying linear transformations of word embeddings seems limited. A comment on the current relevance of the dataset would be helpful.”
>
>     - This is an excellent comment and, indeed, worth discussing. Note that a word is still the foundation of any language, and word embeddings directly impact the performance of downstream NLP tasks. Although contextual embeddings and in-context learning capabilities are the central focus of NLP now, those contextual embeddings are usually derived from nonlinear transformations of context-free (classic) embeddings and hence, improving classic context-free embeddings will directly/indirectly improve contextual embeddings and in-context learning capabilities, which in turn, will also enable improvement in sentence/document-level and downstream tasks.

---

### Official Review · Reviewer_ZKu3 · 2023-08-05

**Soundness:** 2

**Excitement:**

2: Mediocre: This paper makes marginal contributions (vs non-contemporaneous work), so I would rather not see it in the conference.

**Paper Topic And Main Contributions:**

The authors of this paper tried to build a benchmark dataset of Bangala word analogies, in addition to translating the word analogies of another English dataset; Mikolov dataset. They used both datasets in evaluating the quality of existing Bangala word embeddings. Unfortunately, their experiments with different state-of-the-art embedding models reveal that current Bangla word embeddings couldn't achieve high accuracy on both datasets.

**Reasons To Accept:**

The real contribution of this paper is the datasets developed by the authors for the Bangala language which is considered one of the low-resource languages although it is spoken by over 300 million native speakers.

**Reasons To Reject:**

Although the paper presents a good trial for developing two datasets for Bangala, they couldn't achieve high results in their experiments. This may be due to the low representativeness of their compiled and translated dataset or the unbalance of these datasets. In addition, in their experiments, they experiment on each dataset separately, but they can merge the two datasets which may enhance the results. They need to enrich their datasets with more word analogies and try different experiments to achieve best results.


**Reproducibility:**

2: Would be hard pressed to reproduce the results. The contribution depends on data that are simply not available outside the author's institution or consortium; not enough details are provided.

**Reviewer Confidence:**

4: Quite sure. I tried to check the important points carefully. It's unlikely, though conceivable, that I missed something that should affect my ratings.

**Typos Grammar Style And Presentation Improvements:**

Lines 169-173, you need to clarify what you mean by top-1, top-3, top-5, and top-10. This is used several times later and it is not clear.

---

> ### Author Rebuttal · Authors · 2023-08-26
>
> “Although the paper presents a good trial for developing two datasets for Bangala, they couldn't achieve high results in their experiments. This may be due to the low representativeness of their compiled and translated dataset or the unbalance of these datasets. In addition, in their experiments, they experiment on each dataset separately, but they can merge the two datasets which may enhance the results. They need to enrich their datasets with more word analogies and try different experiments to achieve best results.”
>
> - The main goal of our paper is to create a benchmark word analogy dataset (the first of its kind for Bangla) to facilitate future Bangla NLP research and evaluation. Although the results reported in this paper do not show a high degree of performance, we want to emphasize that achieving a high performance is NOT the goal of this work. In fact, achieving high performance is an orthogonal goal, and to achieve this goal, we first need an evaluation benchmark, which does not exist today for Bangla word analogy tasks. To address this issue, we created two different datasets. The first dataset keeps the distinct linguistic characteristics that distinguish Bangla from English and make it unique (see Table 1). The second dataset is a translation of Mikolov's original word analogy dataset that has been meticulously manually filtered to ensure alignment with the Bangla language. Our analyses show that both of these datasets' performance is noticeably limited, indicating significant room for future improvement in Bangla embeddings. However, optimizing for the best results is out of the scope of our paper.
>
>
>
> “you need to clarify what you mean by top-1, top-3, top-5, and top-10. This is used several times later and it is not clear.”
>
> - To clarify, these terms indicate whether the correct answer is present within the top 1, 3, 5, and 10 outputs based on the best cosine similarity scores with other words in the vocabulary.

---

### Meta-Review · Area_Chair_VpdT · 2023-09-17

**Recommendation:** 4

**Metareview:**

In this paper, the authors create a new word analogy dataset for Bangla. A diverse set of relations are covered spanning many synactic and semantic relation categories. They also translate and filter the original Mikolov dataset. They compare various models (classical word embedding models and sentence embeddings models) on this dataset.

Reasons To Accept:
- The datasets developed by the authors for the Bangala language which is considered one of the low-resource languages although it is spoken by over 300 million native speakers.
- Creation of a word analogy dataset for Bangla that is diverse
- Translation of an English word analogy dataset can help cross-lingual studies.
- Various word representation methods are evaluated and compared, including their fine-grained evaluation on the 12 word analogy relationship types.
- The paper is well written and easy to follow.

Reasons To Reject:
- Although the paper presents a good trial for developing two datasets for Bangala, they couldn't achieve high results in their experiments.
- There is limited learning from the paper about word analogy in Bangla beyond reporting accuracies on a few pre-trained embeddings.

I agree with the authors when they write (during the rebuttal): "The main goal of our paper is to create a benchmark word analogy dataset (the first of its kind for Bangla) to facilitate future Bangla NLP research and evaluation. Although the results reported in this paper do not show a high degree of performance, we want to emphasize that achieving a high performance is NOT the goal of this work."

---

### Decision · Program_Chairs · 2023-10-07

**Decision:**

Accept-Main

**Comment:**

In this paper, the authors create a new word analogy dataset for Bangla. A diverse set of relations are covered spanning many synactic and semantic relation categories. They also translate and filter the original Mikolov dataset. They compare various models (classical word embedding models and sentence embeddings models) on this dataset.

Reasons To Accept:
- The datasets developed by the authors for the Bangala language which is considered one of the low-resource languages although it is spoken by over 300 million native speakers.
- Creation of a word analogy dataset for Bangla that is diverse
- Translation of an English word analogy dataset can help cross-lingual studies.
- Various word representation methods are evaluated and compared, including their fine-grained evaluation on the 12 word analogy relationship types.
- The paper is well written and easy to follow.

Reasons To Reject:
- Although the paper presents a good trial for developing two datasets for Bangala, they couldn't achieve high results in their experiments.
- There is limited learning from the paper about word analogy in Bangla beyond reporting accuracies on a few pre-trained embeddings.

I agree with the authors when they write (during the rebuttal): "The main goal of our paper is to create a benchmark word analogy dataset (the first of its kind for Bangla) to facilitate future Bangla NLP research and evaluation. Although the results reported in this paper do not show a high degree of performance, we want to emphasize that achieving a high performance is NOT the goal of this work."